# Does "lottery culture" affect household financial decisions? Evidence from China

Dongmei Cao[1], Dan Wang[2]*, Yujia Liao[1], Qing Liu[1]

**1** School of Public Administration, Southwestern University of Finance and Economics, Chengdu, Sichuan, China, **2** School of Management, University of Science and Technology of China, Hefei, Anhui, China

* wd0319@mail.ustc.edu.cn

## Abstract

In recent years, China's "lottery culture" has developed vigorously. Moreover, the investment participation rate of Chinese families in the formal financial market is low, whereas that in the informal financial market is high. Is there a certain relationship between "lottery culture" and family financial decision-making? If so, what is the underlying mechanism? Based on the 2017 CHFS data and lottery sales data of provinces, this study explores the impact of "lottery culture" on household participation in the formal and informal financial markets and the diversity of household financial portfolios. Results show that "lottery culture" can impede household participation in the formal financial market and the diversity of household financial portfolios while promoting household participation in the informal financial market in China. Furthermore, we analyze two channels of "lottery culture" impacts on household financial decisions: (1) risk attitude and (2) human capital. Results illustrate that "lottery culture" can influence household financial decisions by increasing risk tolerance and reducing the human capital of households.

**Data Availability Statement:** Data cannot be shared publicly because of the data use policy of the Southwestern University of Finance and Economics. Data are available from the Survey and Research Center for China Household Finance

## 1 Introduction

As one of the world's biggest economies, China is now experiencing a period of rapid development in the lottery industry. Data from Chinese Research Data Services show that the sum of lottery sales in China increased by 382% from 2008 to 2018, indicating that gambling attains wider acceptability in Chinese society, and a "lottery culture" emerges [1]. "Lottery culture" is significantly different from the dialect, trust, and other cultural types discussed in previous studies. Buying lottery tickets was once considered a game behavior. That is, under the premise that multiple decision makers interact and exert influence on each other, each decision maker chooses the decision behavior that can bring maximum value to itself according to the information and personal cognitive judgment he/she have [2]. In essence, the lottery meets the individual's psychology of being small and broad, and the accompanying "lottery culture" can be defined as reflecting the high degree of recognition of regional decision-makers for "small probability, high yield" events [3]. Previous studies showed that "lottery culture" has significant impacts on corporate decisions, household investment portfolio choices, and stock returns. Chen et al. [4] and Adhikari et al. [5] used samples of US companies and found that

Institutional Data Access (https://chfs.swufe.edu.
cn/sjzx.htm) for researchers who meet the criteria
for access to confidential data. The data underlying
the results presented in the study are available
from Research Center for China Household Finance
(https://chfser.swufe.edu.cn/datas/).

**Funding:** The author(s) received no specific
funding for this work.

**Competing interests:** The authors have declared
that no competing interests exist.

gambling preference may lead to higher tolerance of failure, thereby promoting corporations' innovation. Kumar et al. [6] explored the impact of "lottery culture" on financial market outcomes. Using the Catholic–Protestant ratio as a proxy of "lottery culture," they found that investors tend to hold more lottery stock and the magnitude of negative lottery stock is larger in regions with higher gambling preferences. Using large administrative Swedish data, Briggs et al. [7] found that windfall gain increases the stock market participation probability. With the development of China's lottery industry, scholars began to pay attention to the influence of "lottery culture" in China. Unlike in the US, the "lottery culture" in China inhibits corporations' innovation [8], and firms in regions with stronger gambling preference experience greater stock price crash risk [9]. However, research providing empirical evidence on the impact of "lottery culture" on household financial decisions in China is limited. The above-mentioned studies motivate us to answer the question: Does "lottery culture" influence household financial decisions in China?

President Xi redefines the principal contradiction in China as the contradiction between unbalanced and inadequate development and the people's ever-growing need for a better life. From the household finance perspective, inadequate development is reflected in the low rate of household formal financial market participation [10] and the lack of diversity in the household financial portfolio that households hold. Unbalanced development is reflected in the differences between the development of the formal and informal financial markets. Approximately 31% of households participate in the stock market in the US [11]. In comparison, the estimated household formal financial market participation rate in China is only 8.6%, and only 9.6% of households in China hold more than one financial asset in the formal financial market based on the data from China Household Finance Survey (CHFS) in 2017. In the meantime, financial depression still exists in China, and some households cannot easily gain formal financial services from banks or securities companies. Therefore, households that cannot have access to formal financial services prefer to participate in the informal financial market. As a major component of the informal financial market in China, private lending usually carries higher interest rates than bank wealth management products. Moreover, based on the data from 2017 CHFS, the proportion of households participating in private lending is 19.2%. However, households participating in private lending may encounter a Ponzi scheme, and high interests are not protected by the law in China. Therefore, the causes of the low rate of formal financial market participation, the high rate of informal financial market participation, and the low diversified financial portfolio are important issues in China.

The existing literature explains the causes of household financial decisions from several aspects including financial availability, transaction costs, household demographic characteristics, and macro environment. Yin et al. [12] found that an increase in financial availability will enable households to increase risky assets held in the formal financial market and decrease that in the informal financial market. Cocco et al. [13] revealed that participating in the financial market is difficult for households with low wealth because of the tax and transaction costs. Bricker et al. [14] found that relatively richer households use more debt as they signal their higher status to their neighbors through the consumption of visible status goods. Rosen and Wu [15] showed that human capital accumulation is one of the key factors that influence household financial decisions. Campbell [16] found that wealth, income, age, race, education, and risk attitude are correlated to household financial decisions. Van Rooij et al. [17] showed that people with low financial literacy are less likely to invest in stocks. Yang et al. [10] illustrated that religious faith can promote household financial market participation and increase risky asset holding. Li et al. [18] examined high sex ratios would raise the household stock market participation rate. Hanspal et al. [19] surveyed a representative sample of US households. They found that exposure to the wealth shocks from the COVID-19 stock market crash

affected household expectations about their wealth, planned investment decisions, and labor market activity. However, few studies paid attention to the impact of "lottery culture" on household financial decisions.

Based on the 2017 CHFS data and lottery sales data of provinces, this study uses the ordinary least squares (OLS), instrumental variables (IV), and mediation models to examine the influence of "lottery culture" on a household financial decision, including possible channels through which "lottery culture" has an influence on household financial decisions. Our results show that "lottery culture" has negative and positive impacts on the depth and probability of households' formal and informal financial market participation, respectively. In addition, "lottery culture" can reduce the diversity of household financial portfolios, which is in line with Shu et al. [20]. Further analysis shows that "lottery culture" influences household financial decisions by affecting the household's risk attitude and human capital.

Our study contributes to the literature in the following ways. First, our study broadens the research field on the causes that lead to limited formal financial market participation and undiversified portfolio choice from a "lottery culture" perspective. To the best of our knowledge, this study is the first to explore the impact of "lottery culture" on household financial decisions in China. Second, this study reveals that risk attitude and human capital accumulation are two factors driving the relation between "lottery culture" and household financial decisions. Although our channel analysis may not completely cover the channels of impacts of "lottery culture" on household financial decisions, we provide possible explanations.

The remainder of the paper is structured as follows. Section 2 presents the literature review and the research hypothesis, and Section 3 presents the data and summary statistics. Next, Section 4 reports the empirical results, and Section 5 conducts a robustness test. Then, Section 6 reports further analysis results. Finally, Section 7 concludes the study.

## 2 Hypothesis development

### 2.1 "Lottery culture" and household participation in the financial market

Local culture shapes local views on wealth and education, which, in turn, influences household financial decisions [10]. As one of the local cultures, "lottery culture" represents residents' gambling preferences. Moreover, even those residents who have not bought lottery tickets, their preferences and behavior are influenced by the opinions of their neighbors, friends, and colleagues who buy lottery tickets.

"Lottery culture" may influence household participation in the formal and informal financial markets in two ways. First, "lottery culture" may influence household participation in the formal and informal financial markets through a household risk attitude. On the one hand, "lottery culture" reflects local gambling preference [8] and represents the risk attitude of households. Spurrier and Blaszczynski [21] explained that investors with gambling preferences have a more optimistic overall perception of risk and prefer to take risky choices. Lo et al. [22] showed that gambling preferences reflect a speculative mentality, and households are willing to take more risks for getting the maximum return quickly with minimum effort. On the other hand, an increase in risk tolerance of households can promote household participation in the formal and informal financial markets. Guiso et al. [23] used consumer survey data from Italian Bank and found that risk averse people are less likely to invest in risky financial assets. Halko et al. [24] explored differences between men and women participation in the stock market and showed that the higher risk aversion of women is the reason that women participate less in the stock market. Vaarmets et al. [25] also showed that people who participate in the stock market are more risk tolerant. The formal and informal financial markets have lottery-type financial assets. Therefore, based on these studies, we can assume that "lottery culture"

increases the risk tolerance of households, thereby promoting household participation in the formal and informal financial markets.

Second, "lottery culture" influences household participation in the formal and informal financial markets through human capital. "Lottery culture" can reduce the human capital of households. Neighbors et al. [26] showed that gambling activities may cause serious health problems, work, and educational disruption. Gambling can also lead to cognitive biases [27]. Furthermore, Yin et al. [12] and Vaarmets et al. [25] found that human capital plays an important role in promoting household participation in the formal financial market. The reason is that investors with high intelligence can understand the benefit and the cost of risky financial projects easily. Therefore, "lottery culture" may reduce household participation in the formal financial market by inhibiting human capital accumulation. However, existing studies showed that although human capital can promote household participation in the formal financial market, it can reduce household participation in the informal financial market. On the one hand, compared with the informal financial market, the formal financial market has a certain investment threshold [28]. For example, investors need more than 500 thousand RMB if they want to invest in options in China. Households with higher human capital are more likely to obtain higher income and thus are more likely to participate in the formal financial market and less likely to participate in the informal financial market [29]. On the other hand, the formal financial market has various financial products, requiring households to have a certain cognitive ability, financial literacy, and math skill to identify risks and possible gains. In the meantime, financial products in the informal financial market in China are very few, mainly private lending and borrowing. Therefore, households with lower human capital are more likely to participate in the informal financial market. Yin et al. [12] showed that the education level of the head of household has a significant positive and negative effect on participation in the formal and informal financial markets, respectively. Based on these studies, we can assume that "lottery culture" reduces human capital, thereby reducing household participation in the formal financial market while promoting that in the informal financial market.

"Lottery culture" can promote and reduce household participation in the formal financial market through the risk attitude channel and human capital channel, respectively. Therefore, determining the net impact of "lottery culture" on formal financial market participation theoretically is difficult. Hence, our hypotheses are as follows:

Hypothesis 1a. "Lottery culture" can promote household participation in the formal and informal financial markets.

Hypothesis 1b. "Lottery culture" can impede household participation in the formal financial market and promote household participation in the informal financial market.

## 2.2 "Lottery culture" and diversity of household financial portfolios

"Lottery culture" may also influence the diversity of household financial portfolios by affecting risk attitude and human capital. Barasinska et al. [30] and Li et al. [31] showed that risk averse households prefer to hold risk-free assets; therefore, an increase in risk tolerance is positively related to the diversity of household financial portfolios. However, Gome and Michaelides [32] found that risk averse household are more likely to hedge financial risk by diversifying their assets, and an increase in risk tolerance is negatively related to the diversity of household financial portfolios. Campbell et al. [33] considered both thoughts above and showed that households with moderate risk aversion are more likely to hedge risk through diversification, whereas extreme risk averse and risk tolerate households prefer to hold a lower diversified financial portfolio. Therefore, based on existing studies, the impact of risk attitude on the diversity of financial portfolios is still mixed.

Households with higher human capital can gather and understand financial information more efficiently. Therefore, they are more capable to understand the advantage of diversity and to reduce risk through diversification. Abreu and Mendes [34] found that household financial knowledge has a significant impact on the number of assets included in a financial portfolio. Using the data of 2013 CHFS, Zeng et al. [35] showed that the higher the level of household financial knowledge, the more diversified the household portfolio.

As the impact of "lottery culture" on the diversity of financial portfolios through the risk attitude channel is still unclear, determining the net impact of "lottery culture" on the diversity of household financial portfolios directly is difficult. Therefore, our hypotheses are as follows:

Hypothesis 2a. "Lottery culture" can promote the diversity of household financial portfolios.

Hypothesis 2b. "Lottery culture" can impede the diversity of household financial portfolios.

## 2.3 Mechanism analysis

As we mentioned in Section 2.1, the impacts of "lottery culture" on household participation in the formal and informal financial markets and the diversity of household financial portfolios may be attributed to risk attitude and human capital. Therefore, our hypotheses are as follows:

Hypothesis 3a. "Lottery culture" influences household financial decisions through risk attitudes.

Hypothesis 3b. "Lottery culture" influences household financial decisions through human capital.

# 3 Model and data description

## 3.1 Model specification

In the baseline model, we use the proportion of risky financial assets in the formal and informal financial markets to financial assets to measure the depth of household participation in both financial markets. Therefore, the OLS model is appropriate for our analysis. The baseline model is set as follows:

$$Y = \alpha LotteryCulture + X\beta + \mu. \tag{1}$$

*Y* represents household participation in the formal and informal financial markets and the diversity of financial portfolios. *LotteryCulture* is the indicator of regional "lottery culture." *X* is a vector of control variables, including the head of household and household characteristics. In addition, China's government classified provinces into four economic zones based on the level of regional economic and social development. Therefore, we also include economic zone fixed effects in the regression to control regional differences.

## 3.2 Data description

Our dataset is combined with the CHFS conducted in 2017, the sum of lottery sales of provinces from the Chinese Research Data Services Platform, and the per capita GDP of provinces from the Easy Professional Superior database. The CHFS survey in 2017 covers 40,011 households in 335 counties in 29 provinces. After deleting the missing items, we have 29,838 observations.

**3.2.1 Explained variables.** The explained variables are household participation in the formal and informal financial markets and the diversity of household financial portfolio. Financial assets include risky assets in the formal financial market, risky assets in the informal financial market, and risk-free financial assets. Risky assets in the formal financial market

include stock, fund, corporate bond, financial bond, financial derivatives, financial wealth management products, foreign exchange, and gold. Risky assets in the informal financial market in China mainly refer to private loans that households lend. Risk-free financial assets include cash, cash in the sock account, government bond, current deposit, and time deposit. Household participation in the formal financial market is defined as the proportion of risky assets in the formal financial market to total financial assets. Household participation in the informal financial market is defined as the proportion of risky assets in the informal financial market to total financial assets. Moreover, the diversity of household financial portfolios is measured as follows:

$$Diversity = \sum_{i=1}^{N} w_i^2, \tag{2}$$

where $N$ represents the number of different financial assets that households have and $w_i$ represents the proportion of financial asset $i$ in the total financial assets that households have. We only contain financial assets in the formal financial market and risk-free financial assets when constructing the index of the diversity of financial portfolios, and we regard risk-free financial assets as one type of asset. We did not include private loans because we cannot regard all private loans as one type of asset. Private loans' risk and return in China vary a lot and are considered a type of asset that can lead to bias. Moreover, we cannot have the specific type of household private loans in the informal market from CHFS.

**3.2.2 Explanatory variable.** The explanatory variable is "lottery culture." Two main types of lottery exist in China: welfare lottery and sports lottery. Referring to Zhao et al. [8] and Christensen et al. [2], we measure the index by 1000 times the ratio of per capita lottery sales to per capita GDP of provinces. Lottery sales are constructed as the sum of welfare lottery and sports lottery sales.

**3.2.3 Control variables.** Our control variables contain the head of household and household characteristics. The head of household is a household member who has the decision-making power. The head of household characteristics include gender, marriage, age, square of age, whether the head of household has medical insurance, whether the head of household has social insurance, health, happiness, and whether the head of household possesses credit cards. The household characteristics comprise household income, household debt, and whether the household locates in rural places. Table 1 presents the variable names, descriptions, and sources.

Table 2 presents the summary statistics. As shown in Table 2, the risky asset in the formal financial market of total financial assets is 4.22%, which is much lower than that in the informal financial market. Moreover, the diversity of household financial portfolios is only 0.0363, which is much lower than 1, indicating that the number of different financial assets held by Chinese households is few.

## 4 Empirical results

To determine the empirical results more clearly in the next parts, this study will use "Formal" to represent the explained variable "Formal Financial Market Participation," "Informal" to represent the explained variable "Informational Financial Market Participation," and "Diversity" to represent the explained variable "Diversity of Financial Portfolio".

First, we examine the impact of "culture lottery" on household participation in the formal and informal financial markets using OLS regressions. Standard errors are clustered at the county level. As shown in column 1 of Table 3, the coefficient estimate of "lottery culture" is negative and significant at the 5% level, indicating that "lottery culture" can significantly reduce household participation in the formal financial market. Column 2 shows that the

Table 1. Variable names, descriptions, and sources.

| Variable name | Variable description | Source |
|---|---|---|
| Formal financial market participation | The proportion of household-owned financial Risky assets in the formal financial market to total financial assets | China Household Finance Survey |
| Informal financial market participation | The proportion of household-owned risky financial assets in the informal financial market to total financial assets | China Household Finance Survey |
| Diversity of financial portfolio | The diversity of financial assets that households hold | China Household Finance Survey |
| Lottery culture | 1000× (per capital lottery sales/per capita GDP) of provinces | Chinese Research Data Services Platform & Easy Professional Superior database |
| Gender | Male = 1, female = 0 | China Household Finance Survey |
| Marriage | Unmarried = 2, Married = 2, Cohabitation = 3, Separated = 4, Divorced = 5, Widowed = 6, Remarried = 7 | China Household Finance Survey |
| Age | The age of the head of household | China Household Finance Survey |
| Income | The natural log of one added to the total household income | China Household Finance Survey |
| Debt | The natural log of one added to the total household debt | China Household Finance Survey |
| Rural | Household located in countryside = 1, otherwise = 0 | China Household Finance Survey |
| Medical insurance | The head of household has medical insurance = 1, otherwise = 0 | China Household Finance Survey |
| Social insurance | The head of household has social insurance = 1, otherwise = 0 | China Household Finance Survey |
| Health | Physical condition of the head of household: very good = 1, good = 2, ordinary = 3, bad = 4, very bad = 5 | China Household Finance Survey |
| Credit card | The head of household has credit cards = 1, otherwise = 0 | China Household Finance Survey |
| Happiness | Very happy = 1, happy = 2, generally = 3, unhappy = 4, very unhappy = 5 | China Household Finance Survey |

coefficient estimate is significantly positive, illustrating that "lottery culture" can promote household participation in the informal financial market. The preliminary findings confirm Hypothesis 1b and reject Hypothesis 2b, indicating that the inhibitory impact of "lottery culture" on household participation in the formal financial market through the human capital channel dominates in China.

Table 2. Summary statistics.

| Variable | Obs | Mean | SD | Min | 10% | 90% | Max |
|---|---|---|---|---|---|---|---|
| Formal financial market participation | 29838 | 0.0422 | 0.1551 | 0.0000 | 0.0000 | 0.0000 | 1.0000 |
| Informal financial market participation | 29838 | 0.0724 | 0.2082 | 0.0000 | 0.0000 | 0.2597 | 1.0000 |
| Diversity of financial portfolio | 29818 | 0.0363 | 0.1230 | 0.0000 | 0.0000 | 0.0000 | 0.7696 |
| Lottery culture | 29948 | 5.3007 | 1.6009 | 2.6363 | 3.8428 | 7.1670 | 10.7139 |
| Gender | 29948 | 0.2028 | 0.4021 | 0.0000 | 0.0000 | 1.0000 | 1.0000 |
| Marriage | 29948 | 2.3933 | 1.2235 | 1.0000 | 2.0000 | 5.0000 | 7.0000 |
| Age | 29948 | 55.2391 | 14.0804 | 3.0000 | 36.0000 | 74.0000 | 117.0000 |
| Income | 29948 | 10.5554 | 1.6796 | 0.0000 | 8.6182 | 12.0624 | 15.4250 |
| Debt | 29948 | 3.3354 | 5.0952 | 0.0000 | 0.0000 | 11.6953 | 15.4250 |
| Rural | 29948 | 0.3185 | 0.4659 | 0.0000 | 0.0000 | 1.0000 | 1.0000 |
| Medical insurance | 29948 | 0.9359 | 0.2450 | 0.0000 | 1.0000 | 1.0000 | 1.0000 |
| Social insurance | 29948 | 0.8264 | 0.3787 | 0.0000 | 0.0000 | 1.0000 | 1.0000 |
| Health | 29945 | 2.6069 | 1.0100 | 1.0000 | 1.0000 | 4.0000 | 5.0000 |
| Credit card | 29843 | 0.8095 | 0.3927 | 0.0000 | 0.0000 | 1.0000 | 1.0000 |
| Happiness | 29928 | 2.1279 | 0.8214 | 1.0000 | 1.0000 | 3.0000 | 5.0000 |

Notes: Obs, Mean, SD, Min, 10%, 90%, and Max represent the total number of observations, mean, standard deviation, minimum, 10% quantile, 90% quantile, and maximum, respectively.

**Table 3. "Lottery culture" and household financial decisions.**

|  | (1) | (2) | (3) | (4) | (5) | (6) |
|---|---|---|---|---|---|---|
|  | Formal | Informal | Diversity | Formal | Informal | Diversity |
|  | OLS | OLS | OLS | IV | IV | IV |
| Lottery culture | −0.0037** | 0.0027** | −0.0030** | −0.0073** | 0.0078** | −0.0089*** |
|  | (−2.4021) | (2.0396) | (−2.5017) | (−1.9819) | (2.2731) | (−2.8565) |
| Gender | YES | YES | YES | YES | YES | YES |
| Marriage | YES | YES | YES | YES | YES | YES |
| Age | YES | YES | YES | YES | YES | YES |
| Age$^2$ | YES | YES | YES | YES | YES | YES |
| Credit card | YES | YES | YES | YES | YES | YES |
| Income | YES | YES | YES | YES | YES | YES |
| Debt | YES | YES | YES | YES | YES | YES |
| Rural | YES | YES | YES | YES | YES | YES |
| Medical insurance | YES | YES | YES | YES | YES | YES |
| Social insurance | YES | YES | YES | YES | YES | YES |
| Health | YES | YES | YES | YES | YES | YES |
| Happiness | YES | YES | YES | YES | YES | YES |
| Constant | YES | YES | YES | YES | YES | YES |
| Area fixed effects | YES | YES | YES | YES | YES | YES |
| First-stage F test | / | / | / | 79.4700 | 79.4700 | 79.4100 |
| R$^2$ | 0.0987 | 0.0431 | 0.1209 | 0.0978 | 0.0421 | 0.1170 |
| Chi | 30.3284 | 61.2343 | 39.6571 | 29.3702 | 60.2483 | 38.4060 |
| Observation | 29713 | 29713 | 29691 | 29713 | 29713 | 29691 |

Notes: Significance levels are denoted as ***$p < 1\%$, **$p < 5\%$, and *$p < 10\%$. Standard errors in parentheses are clustered heteroskedasticity-robust standard errors.

Second, we explore the impact of "culture lottery" on the diversity of household financial portfolios. As shown in column 3 of Table 3, the coefficient estimate of "lottery culture" is negative and significant at the 5% level, which is consistent with hypothesis 2b, supporting that "lottery culture" can impede the diversity of household financial portfolios.

We also use IV estimation to alleviate endogenous problems. Columns 4–6 show the results of IV estimation, where the number of Chinese Football Super League and China Basketball Association teams in 2017 in provinces is used as the instrument variable [2, 8]. Chinese Football Super League and China Basketball Association are the two most important sports events in China. People in provinces that have one of these teams are more likely to be attracted by these events, influencing the sales of local sports lottery. The results of columns 4–6 are consistent with Hypotheses 1b and 2b. The results of the Kleibergen-Paap rk LM statistic and the F test of first-stage regression show that the instrument variable is relevant to "lottery culture" and has no weak instrument variable problems. Whether in theory or reality, sports events do not affect family financial market participation, which is sufficient to show the exogenous nature of IV. To solve the endogenous problem more rigorously, we conducted the second-stage regression and regress the residual to IV. The p value is greater than 0.1, the original hypothesis is not rejected, and then, IV is exogenous.

## 5 Robustness tests

In this subsection, we use binary choice variables to measure the probability of household participation in the formal and informal financial markets. Household participation in the formal

**Table 4. "Lottery culture" and household financial decisions (alternative proxies of explained variables).**

|  | (1) | (2) | (3) |
|---|---|---|---|
|  | Formal | Informal | Diversity |
|  | Logit | Logit | OLS |
| Lottery culture | −0.0794** | 0.0496*** | −0.0113*** |
|  | (−2.0056) | (2.8240) | (−2.7347) |
| Gender | YES | YES | YES |
| Marriage | YES | YES | YES |
| Age | YES | YES | YES |
| Age$^2$ | YES | YES | YES |
| Credit card | YES | YES | YES |
| Income | YES | YES | YES |
| Debt | YES | YES | YES |
| Rural | YES | YES | YES |
| Medical insurance | YES | YES | YES |
| Social insurance | YES | YES | YES |
| Health | YES | YES | YES |
| Happiness | YES | YES | YES |
| Constant | YES | YES | YES |
| Area fixed effects | YES | YES | YES |
| R$^2$ | 0.2489 | 0.0892 | 0.1273 |
| Chi | 2156.8600 | 1912.3800 | 35.6513 |
| Observation | 29820 | 29820 | 29660 |

*Notes*: Significance levels are denoted as ***p < 1%, **p < 5%, *p < 10%. Standard errors in parentheses are clustered heteroskedasticity-robust standard errors.

and informal financial markets is defined in terms of whether the households hold risky assets in the formal and informal financial markets, respectively, which equals 1 if households have risky assets in the formal and informal financial markets, and 0 if otherwise. In addition, we use categories of financial assets that households have to measure the diversity of household financial portfolios. If the household financial investment portfolio includes bonds, stocks, and investment funds, the diversity of household financial assets is assigned a value of 3. Table 4 shows the results.

The observations in Table 4 are larger than baseline samples because even if the household does not report the value of specific financial assets, we can still know whether they participate in the financial market through their answer to the type of asset holdings in the 2017 CHFS. The results in Table 4 show that "lottery culture" reduce the probability of household participation in the informal financial market and the diversity of household financial portfolio. In addition, "lottery culture" increases the probability of household participation in the formal financial market. The results indicate that our main conclusions are robust.

## 6 Further analyses

As we mentioned in Section 2, the impacts of "lottery culture" on household participation in the formal financial market, informal financial market, and the diversity of household financial portfolio may be attributed to risk attitude and human capital. Therefore, we further analyzed the mediation effect of risk attitude and human capital to prove these possible channels in this section.

Following Baron and Kenny [36], we use the mediation model to explore the above channels:

$$M = \chi LotteryCulture + \delta X + \varepsilon, \tag{3}$$

$$Y = \phi M + \varphi LotteryCulture + \kappa X + \varepsilon. \tag{4}$$

$M$ is household risk attitude or human capital. The results of model (1) prove that "lottery culture" can impact household financial decisions. If the results of model (3) can prove that "lottery culture" have an impact on household risk attitude (human capital), and the results of model (4) can prove that household risk attitude (human capital) has an impact on household financial decisions, then we can testify these possible channels.

Household risk attitude is measured by the answer to the question "Which investment project do you prefer to choose if you have a fund for investment?" which equals 1 if households choose "do not want to take any risks" or "slightly lower risk, slightly lower return projects," equals 2 if households choose "projects with average risk and average return", and equals 3 if households choose "high risk, high return project" or "slightly higher risk, slightly higher return projects." The larger the index, the more risk tolerant the household. Household human capital is measured by the education level of the head of household. The larger the index, the higher the education level of the head of household.

Table 5 measures whether "lottery culture" can impact household financial decisions through household risk attitude. Based on the existing studies, household risk attitude may have non-linear impacts on the diversity of the household financial portfolio. Therefore, we add a square of risk attitude when the explained variable is the diversity of household financial portfolios. Column 1 of Table 5 shows that "lottery culture" can increase risk tolerance of households. Moreover, the coefficient estimates of risk attitude are significantly positive in columns 2 and 3, indicating that the increase of household risk tolerance can promote household participation in the formal and informal financial markets. The results are consistent with our analysis in Section 2. In addition, the coefficient estimate of risk attitude is significantly positive, and the coefficient of the square of risk attitude is not significant in column 4. This result shows that the increase of household risk tolerance only has a positive effect on the diversity of household financial portfolio. The results in Table 5 testify that "lottery culture" can influence household financial decisions through risk attitude channel.

Table 6 measures whether "lottery culture" can influence household financial decisions through household human capital. Column 1 of Table 6 shows that "lottery culture" can reduce household human capital. The coefficient estimate of human capital in columns 2 and 3 is significantly positive and significantly negative, respectively. This result shows that the increase of human capital can promote household participation in the formal financial market and impede household participation in the informal financial market. In addition, the coefficient estimate of human capital is positive in column 4, indicating that human capital can increase the diversity of household financial portfolios. The results in Table 6 testify that "lottery culture" can influence household financial decisions through the human capital channel.

## 7 Conclusions

Based on the 2017 CHFS data and lottery sales data of provinces, this study explores the impacts of "lottery culture" on household participation in the formal and informal financial markets and the diversity of household financial portfolios. In addition, this study examines the mediation effects of risk attitude and human capital on the relationship between "lottery culture" and household financial decisions.

Table 5. "Lottery culture," risk attitude, and household financial decisions.

| | (1) | (2) | (3) | (4) |
|---|---|---|---|---|
| | Risk attitude | Formal | Informal | Diversity |
| Risk attitude | | 0.0303*** | 0.0066*** | 0.0193* |
| | | (12.0757) | (2.8561) | (1.9533) |
| Risk attitude$^2$ | | | | 0.0021 |
| | | | | (0.7904) |
| Lottery culture | 0.0075* | −0.0042** | 0.0026* | −0.0035*** |
| | (1.6666) | (−2.5103) | (1.7788) | (−2.7288) |
| Gender | YES | YES | YES | YES |
| Marriage | YES | YES | YES | YES |
| Age | YES | YES | YES | YES |
| Age$^2$ | YES | YES | YES | YES |
| Credit card | YES | YES | YES | YES |
| Income | YES | YES | YES | YES |
| Debt | YES | YES | YES | YES |
| Rural | YES | YES | YES | YES |
| Medical insurance | YES | YES | YES | YES |
| Social insurance | YES | YES | YES | YES |
| Health | YES | YES | YES | YES |
| Happiness | YES | YES | YES | YES |
| Constant | YES | YES | YES | YES |
| Area fixed effects | YES | YES | YES | YES |
| R$^2$ | 0.0998 | 0.1148 | 0.0432 | 0.1405 |
| Chi | 103.8882 | 31.1966 | 55.8685 | 40.0795 |
| Observation | 26157 | 26066 | 26066 | 26049 |

Notes: Significance levels are denoted as ***p < 1%, **p < 5%, and *p < 10%. Standard errors in parentheses are clustered heteroskedasticity-robust standard errors.

Our results show that "lottery culture" impedes household participation in the formal financial market and the diversity of household financial portfolios. In addition, "lottery culture" can promote household participation in the informal financial market. The results are robust when we use IV estimation and alternative proxies of explained variables. Regarding the channels through which "lottery culture" influences household financial decisions, we find that "lottery culture" can affect household financial decisions through household risk attitude and human capital. "Lottery culture" can increase the risk tolerance of households and reduce household human capital, which are key factors that affect household financial decisions.

This study complements the existing literature on local culture and household financial decisions and supplies a possible explanation for limited formal financial market participation and low diversified financial portfolio in China from the "lottery culture" perspective.

Therefore, in the "lottery culture" era in China, the government should balance the development of formal and informal financial markets. The government should take measures to guide people to establish correct financial values, reduce family speculation and game in financial decision-making, and avoid family financial management being excessively biased toward the informal financial market. Moreover, the government should improve the supervision of the formal financial market and enhance people's confidence in the formal financial market so that more people can make correct judgments on the family portfolio.

**Table 6. "Lottery culture," human capital, and household financial decisions.**

|  | (1) | (2) | (3) | (4) |
|---|---|---|---|---|
|  | Education | Formal | Informal | Diversity |
| Education |  | 0.0188*** | −0.0050*** | 0.0155*** |
|  |  | (15.4932) | (−5.3664) | (18.5126) |
| Lottery culture | −0.0344* | −0.0031** | 0.0026* | −0.0025** |
|  | (−1.8413) | (−2.2134) | (1.9404) | (−2.3076) |
| Gender | YES | YES | YES | YES |
| Marriage | YES | YES | YES | YES |
| Age | YES | YES | YES | YES |
| $Age^2$ | YES | YES | YES | YES |
| Credit card | YES | YES | YES | YES |
| Income | YES | YES | YES | YES |
| Debt | YES | YES | YES | YES |
| Rural | YES | YES | YES | YES |
| Medical insurance | YES | YES | YES | YES |
| Social insurance | YES | YES | YES | YES |
| Health | YES | YES | YES | YES |
| Happiness | YES | YES | YES | YES |
| Constant | YES | YES | YES | YES |
| Area fixed effects | YES | YES | YES | YES |
| $R^2$ | 0.3757 | 0.1237 | 0.0441 | 0.1481 |
| Chi | 234.5672 | 34.8319 | 58.0601 | 48.5263 |
| Observation | 29820 | 29713 | 29713 | 29691 |

Notes: Significance levels are denoted as ***$p < 1\%$, **$p < 5\%$, and *$p < 10\%$. Standard errors in parentheses are clustered heteroskedasticity-robust standard errors.

## Author Contributions

**Conceptualization:** Dongmei Cao.

**Data curation:** Dongmei Cao.

**Formal analysis:** Dongmei Cao, Yujia Liao.

**Funding acquisition:** Dongmei Cao.

**Investigation:** Dongmei Cao.

**Methodology:** Dongmei Cao.

**Project administration:** Dongmei Cao.

**Resources:** Dongmei Cao.

**Software:** Dongmei Cao.

**Supervision:** Dongmei Cao.

**Validation:** Dongmei Cao.

**Visualization:** Dongmei Cao.

**Writing – original draft:** Dongmei Cao, Dan Wang, Yujia Liao, Qing Liu.

**Writing – review & editing:** Dongmei Cao, Dan Wang, Yujia Liao, Qing Liu.

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
