## [Decision Letter · Decision Letter 0]

1 Aug 2022

PONE-D-22-15747Does “Lottery Culture” Affect Household Financial Decisions? Evidence from ChinaPLOS ONE

Dear Dr. Wang,

Thank you for submitting your manuscript to PLOS ONE. After careful consideration, we feel that it has merit but does not fully meet PLOS ONE’s publication criteria as it currently stands. Therefore, we invite you to submit a revised version of the manuscript that addresses the points raised during the review process.

We look forward to receiving your revised manuscript.

Kind regards,

Ricky Chee Jiun Chia

Academic Editor

PLOS ONE

Journal Requirements:

Reviewers' comments:

Reviewer's Responses to Questions

**Comments to the Author**

1. Is the manuscript technically sound, and do the data support the conclusions?

Reviewer #1: Partly

Reviewer #2: Yes

2. Has the statistical analysis been performed appropriately and rigorously? 

Reviewer #1: No

Reviewer #2: Yes

3. Have the authors made all data underlying the findings in their manuscript fully available?

Reviewer #1: Yes

Reviewer #2: Yes

4. Is the manuscript presented in an intelligible fashion and written in standard English?

Reviewer #1: Yes

Reviewer #2: Yes

5. Review Comments to the Author

Reviewer #1: Review comments: major revision

Question 1: Can the "lottery culture" of each province from CNRDS be specific to families? Can it be consistent with the family data of CHFS? Please the authors give the answer.

Suggestion 1: This paper uses cross-sectional data, so there should be no “area fixed effects” (because this paper is not panel data). Therefore, it is suggested that the author revalidate the empirical results and remove the line "Area fixed effects" to avoid common-sense empirical errors.

Suggestion 2: It is suggested to add “variable introduction” in this paper, listing “(1) explained variables; (2) Core explanatory variables; (3) Control variables” for more clarity, and should not be mixed into model expressions and data introductions, which appear to be chaotic.

Suggestion 3: The third explained variable —— “the diversity of financial portfolio”. The measurement formula is wrong. It is believed that it should not be “ ”, but should be “ ”，So this is the financial portfolio diversity index..

Suggestion 4: It is suggested that the three explained variables in this paper should be abbreviated in the introduction of variables, so as to be consistent with the empirical results in the following article and not make people feel very abrupt.

Suggestion 5: It is suggested that all control variables in the empirical results do not need to be analyzed by empirical results interpretation. In addition, specific regression coefficients and standard errors are not required to be listed in the table of empirical results, and “YES” can be used to represent control, so as to highlight the influence of the core explanatory variables on the explained variables and echo the theme.

Suggestion 6: In the explanation of endogeneity, it is suggested to explain in detail the two basic conditions —— exogeneity and correlation conditions —— for the selected instrumental variables (because this paper only explains correlation, not exogeneity);For the instrumental variables method used, the test for weak instrumental variables and the test for exogeneity of instrumental variables (only the weak instrumental variables test is explained in this paper) should be performed, and it is suggested that the authors conduct additional tests to ensure the completeness of the endogeneity explanation in this paper.

Suggestion 7: Robustness test: For the first two explained variables, this paper uses the logit model to replace the OLS model. For the third explained variable —— the diversity of financial portfolio. Please explain clearly what specific variable is used to replace the third explained variable in this paper (Table 4,Column 5), and suggest that columns 3-4 of Table 4 should be dropped because they have nothing to do with the robustness check for this article.

Suggestion 8: In the further analysis, that is, the mechanism test in this paper, the random disturbance terms in equations (3) and (4) should be represented by different letters to distinguish them accordingly; In addition, there is no need to list specific regression coefficients and standard errors for all control variables in the table, and “YES” can be used to represent control.

Suggestion 9: It is suggested to add the corresponding research hypothesis of mechanism analysis into the research hypothesis (Hypothesis 3a: "Lottery culture" influences household financial decisions through household risk attitude "; Hypothesis 3b: "Lottery culture" influences household financial decisions through human capital.) In order to echo before and after the part.

Suggestion 10: Pay attention to the translation errors in the text (e.g., Does "Lottery Culture" Affect Household Financial Decisions?—— Evidence from China; Previous research show that "lottery culture" have significant impacts on corporate decisions, Household investment portfolio choices and stock returns. Chen et al.[2], etc.) and standardize the reference format of this paper.

Reviewer #2: (1) What's your definition of “lottery culture”? Can it comprehensively be measured by the ratio of the per capita lottery sales to the per capita GDP of provinces? My suggestion is that the authors should cite more literature.

(2) Is the result of analysis with cross-sectional data convincing? Why didn’t you use panel data? Please explain why you used cross-sectional data instead of panel data.

(3) There are many factors that affect family financial decision-making. Please explain the selection of control variables and the possible reasons for the analysis results.

(4) Is there a science to choosing the number of Chinese Football Super League and China Basketball Association teams as the instrument variable? Please explain it by citing more literature.

(5) Is there any theoretical support for the intermediary mechanism? Why didn’t you propose the hypotheses on it in part 2 (Hypotheses Development)? My suggestion is that the authors should add some description about mediated relation between “lottery culture” and family financial decision-making in the second part.

(6) Is the conclusion of the data analysis in 2017 applicable after 2020 (COVID-19)?

6. PLOS authors have the option to publish the peer review history of their article (what does this mean?). If published, this will include your full peer review and any attached files.

Reviewer #1: **Yes: **Dong Haisong

Reviewer #2: No

---

## [Author Response · Author response to Decision Letter 0]

19 Sep 2022

Response to Reviewer #1

Review comments: major revision

Question 1: Can the "lottery culture" of each province from CNRDS be specific to families? Can it be consistent with the family data of CHFS? Please the authors give the answer.

Response: Thank you for pointing this out to us. This paper refers to the practice of Zhao et al. and measures the variable of “lottery culture” by the index obtained from the ratio of per capita lottery sales to per capita GDP of each province. This is an index that represents the “lottery culture” of a province. This paper matches this index with the location of the family in the CHFS database. If the family belongs to this province, then the “lottery culture index” of this province is used to represent the “lottery culture” of the family. We initially intended to achieve city-level data, but as the relevant data on lottery sales at the city level are not published, calculating the lottery culture index at the city level is impossible.

References: 

Zhao Q, Zhao W, Lu D, Zhao Q. Lottery and enterprise innovation: A research based on the view of culture. Finance & Trade Economics. 2018; 39:122–40.

Christensen D M, Jones K L, Kenchinton D G. Gambling attitudes and financial misreporting. Contemporary Accounting Research,2018,35(3) ;1229—1261

Suggestion 1: This paper uses cross-sectional data, so there should be no “area fixed effects” (because this paper is not panel data). Therefore, it is suggested that the author revalidate the empirical results and remove the line "Area fixed effects" to avoid common-sense empirical errors.

Response: Thank you for this suggestion. In the published journals, research on controlling the fixed effects of industries, cities, provinces, and economic regions in cross-sectional data is not limited. For example, Sun Weizeng’s article “Air pollution and spatial mobility of labor” was published in Economic Research Journal in 2019. In the research design (Y: whether the labor force flows; X: PM2.5), the urban fixed effect is further increased based on the control variables, which alleviates the endogenous problem that may be caused by the omission of urban control variables, obtaining a more robust estimation result.

The “Area fixed effects” used in this paper and the region where the explanatory variables are interpreted are not the same concept. Here, we use the “eastern, northeast, central, and western regions.” The Chinese government divides the provinces into four economic regions according to the level of regional economic and social development: eastern, northeast, central, and western regions. Therefore, this paper also includes the fixed effects of the economic zone into the regression to control regional differences, alleviate the endogenous problems that may be caused by the omission of regional control variables, and obtain more robust estimation results.

Suggestion 2: It is suggested to add “variable introduction” in this paper, listing “(1) explained variables; (2) Core explanatory variables; (3) Control variables” for more clarity, and should not be mixed into model expressions and data introductions, which appear to be chaotic.

Response: Thank you for this suggestion. In the “3.2 Data Description” section, the titles “3.2.1 interpreted variable,” “3.2.2 core explanatory variable,” and “3.2.3 control variable” are added for clarity.

Suggestion 3: The third explained variable —— “the diversity of financial portfolio”. The measurement formula is wrong. It is believed that it should not be “ ”, but should be “ ”，So this is the financial portfolio diversity index.

Response: We first apologize for this mistake. In the current version of the manuscript, we have corrected it after consulting the literature. Thank you for pointing out this.

Suggestion 4: It is suggested that the three explained variables in this paper should be abbreviated in the introduction of variables, so as to be consistent with the empirical results in the following article and not make people feel very abrupt.

Response: Thank you very much for sharing this point. To determine the empirical results more clearly in the next parts, this paper will use “Formal” to represent the explained variable “Formal Financial Market Participation,” “Informal” to represent the explained variable “Informational Financial Market Participation,” and “Diversity” to represent the explained variable “Diversity of Financial Portfolio.”

Suggestion 5: It is suggested that all control variables in the empirical results do not need to be analyzed by empirical results interpretation. In addition, specific regression coefficients and standard errors are not required to be listed in the table of empirical results, and “YES” can be used to represent control, so as to highlight the influence of the core explanatory variables on the explained variables and echo the theme.

Response: Thank you for this great suggestion. According to the suggestion, the control variable is represented by YES, and the interpretation and analysis of the empirical results of all control variables are deleted.

Suggestion 6: In the explanation of endogeneity, it is suggested to explain in detail the two basic conditions —— exogeneity and correlation conditions —— for the selected instrumental variables (because this paper only explains correlation, not exogeneity);For the instrumental variables method used, the test for weak instrumental variables and the test for exogeneity of instrumental variables (only the weak instrumental variables test is explained in this paper) should be performed, and it is suggested that the authors conduct additional tests to ensure the completeness of the endogeneity explanation in this paper.

Response: Thank you for this great suggestion. This supplement is made in the paper: Whether in theory or reality, sports events do not affect family financial market participation, which is sufficient to show the exogenous nature of instrumental variables. To solve the endogenous problem more rigorously, we conducted the second-stage regression and regress the residual to IV. The p value is greater than 0.1, the original hypothesis is not rejected, and then, IV is exogenous.

Suggestion 7: Robustness test: For the first two explained variables, this paper uses the logit model to replace the OLS model. For the third explained variable —— the diversity of financial portfolio. Please explain clearly what specific variable is used to replace the third explained variable in this paper (Table 4, Column 5), and suggest that columns 3-4 of Table 4 should be dropped because they have nothing to do with the robustness check for this article.

Response: Thank you for this suggestion. This paper uses the category of household financial assets to measure the diversity of household financial investment portfolios. For example, a household financial investment portfolio includes bonds, stocks, and investment funds. The category of household financial assets is assigned a value of 3 and so on. Table 4 has columns 3–4 to compare the robustness test results with the original empirical results, but there is no problem in deleting them. We deleted these columns following the reviewer’s comments.

Suggestion 8: In the further analysis, that is, the mechanism test in this paper, the random disturbance terms in equations (3) and (4) should be represented by different letters to distinguish them accordingly; In addition, there is no need to list specific regression coefficients and standard errors for all control variables in the table, and “YES” can be used to represent control.

Response: Thank you for this suggestion. According to the comments, we used ε and replaced μ; the control variable is represented by YES.

Suggestion 9: It is suggested to add the corresponding research hypothesis of mechanism analysis into the research hypothesis (Hypothesis 3a: "Lottery culture" influences household financial decisions through household risk attitude "; Hypothesis 3b: "Lottery culture" influences household financial decisions through human capital.) In order to echo before and after the part.

Response: Thank you for this suggestion. 3.2. Mechanism analysis has been supplemented.

Suggestion 10: Pay attention to the translation errors in the text (e.g., Does "Lottery Culture" Affect Household Financial Decisions?—— Evidence from China; Previous research show that "lottery culture" have significant impacts on corporate decisions, Household investment portfolio choices and stock returns. Chen et al.[2], etc.) and standardize the reference format of this paper.

Response: Thank you for this suggestion. The grammar and translation of this paper have been revised again, and the reference format has been modified according to the standard format.

Once again thank you very much for these great comments.

Response to Reviewer #2

The authors analyze the relationship between “lottery culture” and family financial decision-making through cross-sectional data, and make an analysis of the intermediary mechanism, which is well organized.

Response: Thank you for your appreciation. We truly did our best to improve the manuscript based on your comments.

However, some issues still need to be improved:

(1) What's your definition of “lottery culture”? Can it comprehensively be measured by the ratio of the per capita lottery sales to the per capita GDP of provinces? My suggestion is that the authors should cite more literature.

Response: Thank you very much for sharing these points. This paper has added references to the definition of concepts and variable measures.

After consulting the literature, this paper defines “lottery culture” as reflecting the high degree of recognition of regional decision-makers for “small probability, high yield” events. I made a strengthened explanation in the paper.

At present, most of the papers on lottery culture measured the variable of “lottery culture” using the index obtained from the ratio of per capita lottery sales to per capita GDP of each province, such as Zhao et al. and Christensen et al.

References:

Zhao Q, Zhao W, Lu D, Zhao Q. Lottery and enterprise innovation: A research based on the view of culture. Finance & Trade Economics. 2018; 39:122–40.

Christensen D M，Jones K L, Kenchinton D G. Gambling attitudes and financial misreporting. Contemporary Accounting Research,2018,35(3) ;1229—1261

(2) Is the result of analysis with cross-sectional data convincing? Why didn’t you use panel data? Please explain why you used cross-sectional data instead of panel data.

Response: Thank you for pointing this out to us. CHFS is a sampling survey project carried out by the China Household Finance Research and Survey Center every year across China. It is a general sampling survey, not a tracking survey. The survey items change every year, making it difficult to form panel data with a unified caliber. In addition, the CHFS survey carried out in 2017 covered 40,011 households in 335 counties in 29 provinces. After deleting the missing items, we had 29,838 observations. The sample size is very large, so the analysis results of cross-section are also representative.

(3) There are many factors that affect family financial decision-making. Please explain the selection of control variables and the possible reasons for the analysis results.

Response: Thank you for this great suggestion. In the “3.2 Data Description” section, we introduce the selection of control variables and the introduction of control variables in detail. Under the guidance of your valuable opinions, I have made some improvements.

The details are as follows:

Our control variables contain the head of household and household characteristics. Previous studies showed that the head of household and family characteristics will have an important impact on family financial decisions. The head of household is a household member who has the decision-making power. The head of household characteristics include gender, marriage, age, square of age, whether the head of household has medical insurance, whether the head of household has social insurance, health, happiness, and whether the head of household possesses credit cards. The household characteristics contain household income, household debt, and whether the household is located in rural places. Table 1 documents the variable names, descriptions, and sources.

For the result interpretation of control variables, I originally wrote it in “4. Imperial Results.” However, another reviewer suggested deleting this part because he/she believed that control variables were not the main research point of the paper, and control variables should be represented by YES in the empirical results. I temporarily adopted this suggestion. If necessary, I can restore all the contents of the control variables next time.

This is the result interpretation of my control variable: People with social insurance are more likely to participate in the formal financial market and hold more kinds of financial assets but less likely to participate in the informal financial market. As the age grows, people are more willing to participate in the formal financial market and hold more types of assets. However, when the age exceeds a certain threshold, people change their behaviors. People who have credit cards or are in bad health condition are less likely to participate in the financial market and hold less types of assets. People with high income prefer to participate in the financial market and hold diversified financial portfolios, whereas those who have debt prefer to participate in the informal financial market and are less likely to participate in the formal financial market and hold less diversified financial portfolios. Living in a rural region and having happy feelings can impede household participation in the financial market.

(4) Is there a science to choosing the number of Chinese Football Super League and China Basketball Association teams as the instrument variable? Please explain it by citing more literature.

Response: Thank you for this suggestion.The selection of instrumental variables in this paper refers to the following two studies:

Zhao Q, Zhao W, Lu D, Zhao Q. Lottery and enterprise innovation: A research based on the view of culture. Finance & Trade Economics. 2018; 39:122–40.

Christensen D M, Jones K L, Kenchinton D G. Gambling attitudes and financial misreporting. Contemporary Accounting Research,2018,35(3) ;1229—1261.

(5) Is there any theoretical support for the intermediary mechanism? Why didn’t you propose the hypotheses on it in part 2 (Hypotheses Development)? My suggestion is that the authors should add some description about mediated relation between “lottery culture” and family financial decision-making in the second part. 

Response: Thank you for this suggestion. According to the comments, we have supplemented “2.3 Mechanism analysis” and proposed “Hypothesis 3a. ‘Lottery culture’ influences household financial decisions through risk attitudes” and “Hypothesis 3b. ‘Lottery culture’ influences household financial decisions through human capital.”

Notably, in the “2.1 ‘Lottery Culture’ and Household Participation in The Financial Market,” section, I analyze the theoretical mechanism to determine the causal relationship between “Lottery Culture” and “Household Participation in The Financial Market.” Therefore, Section 2.3 does not discuss too much.

(6) Is the conclusion of the data analysis in 2017 applicable after 2020 (COVID-19)?

Response: Thank you for giving me a very good research entry point. Under the background of the new crown epidemic, what is the impact of lottery culture on family financial decision-making? Can lottery culture even affect family financial decisions? After the COVID-19 epidemic, some systematic risks have emerged in China’s financial market. The risks of investment and financial management are unprecedented. Do families still have money to invest in the financial market? If so, are they willing to invest in the financial market? This point may be more complex and interesting than the article I am currently discussing. I will go to search and see if there are such data to support this study. I believe it will be very meaningful research.

Once again thank you very much for these great comments.

---

## [Editor Report · Decision Letter 1]

23 Sep 2022

Does “lottery culture” affect household financial decisions? Evidence from China

PONE-D-22-15747R1

Dear Dr. Dan Wang,

We’re pleased to inform you that your manuscript has been judged scientifically suitable for publication and will be formally accepted for publication once it meets all outstanding technical requirements.

Kind regards,

Ricky Chee Jiun Chia

Academic Editor

PLOS ONE
---

## [Editor Report · Acceptance letter]

6 Oct 2022

PONE-D-22-15747R1 

Does “lottery culture” affect household financial decisions? Evidence from China 

Dear Dr. Wang:

I'm pleased to inform you that your manuscript has been deemed suitable for publication in PLOS ONE. Congratulations! Your manuscript is now with our production department. 

Kind regards, 

on behalf of

Dr. Ricky Chee Jiun Chia 

Academic Editor

PLOS ONE